# Generalization of fear learning is shaped by inhibitory sensory processing in mice

Alper K. Bakir [1] ✉, Ankur A. Gaikwad [1], Natalia Efimova[1], Katharina L. L. Clark[1], Michelle C. Rosenthal[1] & John P. McGann [1,2,3] ✉

When an organism learns that a sensory stimulus predicts a threat, the brain's neural representation of the stimulus somehow incorporates that information into early sensory processing. That altered sensory processing may causally shape the organism's response to the stimulus. We used contextual fear conditioning in mice to induce odor avoidance that generalized across odors in proportion to their similarity to the conditioning odor. Visualization of odor representations in vivo revealed they were being reshaped by reductions in $GABA_B$ receptor-mediated inhibitory signaling in the presynaptic terminals of the olfactory nerve and downstream neurons in proportion to the generalized fear of each odor. Locally blocking $GABA_B$ receptors in the olfactory bulb caused mice to switch from generalizing fear in proportion to similarity to instead overgeneralizing fear to all odors equally. Learning-induced sensory plasticity thus plays a causal role in shaping fear generalization.

Fear is adaptive when it is correctly matched to actually or potentially threatening stimuli[1–3]. However, learned fear can also overgeneralize to situations that do not predict a threat, driving a range of maladaptive sensory, emotional, cognitive, and behavioral outcomes[1,4–10]. Both stimulus-specific and generalizing fear learning alters the neurosensory representations of potentially threatening sensory stimuli[11–15], sometimes very early in sensory processing[16–19]. While this sensory neuroplasticity is usually interpreted in the context of fear-related perceptual changes[20–22], it remains unclear whether fear-induced sensory neuroplasticity plays a causal role in coupling the sensory stimulus to fear and fear-related behavior. Because fear generalization often occurs along sensory dimensions[7,23,24], such a role could have important implications for the etiology and maintenance of anxiety disorders[2,5].

In the olfactory system, pairing an odor with a shock induces extensive neuroplasticity and perceptual changes in both humans[20,25–27] and mouse models[28,29]. These changes manifest at every level, including the olfactory sensory neurons (OSNs)[16,30,31], periglomerular (PG) and mitral cells in the olfactory bulb[29,32], and downstream cortices[14,33]. When mice learn that a specific odor predicts a shock, the odor-evoked OSN output from the nose to the brain becomes larger

and more distinct from other odors[30], while if the mouse generalizes its fear to other odors then the OSN response is elevated for all those odors[34] and the representations become more similar[32]. It remains unknown how these neurobiological changes occur or whether they play a causal role in odor-evoked fear expression or fear generalization.

OSN axons compose the olfactory nerve, which innervates the surface of the brain's olfactory bulb in discrete glomeruli corresponding to each olfactory receptor[35]. Neurotransmitter release from these OSN synaptic terminals is modulated by presynaptic $GABA_B$ receptors[36–39], which are tonically activated by spillover GABA release from local periglomerular interneurons[40–42] and rapidly modulate conductance through N-type calcium channels[38]. Fear learning downregulates $GABA_B$ receptor expression in OSNs stimulated prior to a footshock[31], suggesting that relief of $GABA_B$ receptor-mediated inhibition of OSN outputs could play a role in instantiating olfactory fear learning and fear generalization by determining which olfactory bulb glomeruli receive fear-potentiated peripheral input. $GABA_B$ receptor-mediated signaling also shapes the activity of multiple other cell types in the olfactory bulb, including periglomerular interneurons, external tufted cells, and mitral cells, suggesting that $GABA_B$ receptor signaling

[1]Behavioral and Systems Neuroscience, Psychology Department, Rutgers, The State University of New Jersey, Piscataway, NJ, US. [2]Rutgers Center for Cognitive Science, Rutgers, The State University of New Jersey, Piscataway, NJ, US. [3]Department of Otolaryngology-Head and Neck Surgery, Rutgers Robert Wood Johnson Medical School, New Brunswick, NJ, US. ✉e-mail: alperkaganbakir@gmail.com; john.mcgann@rutgers.edu

could potentially impact not only primary sensory inputs to the bulb but also the local circuitry that shapes responses within and across odorants.

Within the olfactory bulb, different circuits modulate OSN inputs within and between glomeruli, providing a potential substrate for receptor-specific and generalized gain control, respectively[22,40]. Disinhibition of local inhibitory circuitry during learning is a common motif during associative learning[43,44], including in sensory brain regions. In the auditory cortex, for example, fear learning disinhibits a local microcircuit, thus amplifying pyramidal cell responses to the threat-predictive stimulus that are necessary for learning[45]. Here we test whether odor-cued fear learning can reduce GABA$_B$ receptor-mediated inhibition of the peripheral olfactory input to the brain's olfactory bulb and whether local modulation of this inhibition can cause the generalization of conditioned fear.

## Results

To test this hypothesis, we trained mice on a generalizing fear conditioning task comprised of three daily conditioning sessions in a chamber scented with the odorant methyl valerate (MV, an ester) where they received 10 mild, randomly timed footshocks over the course of about 10 min (Fig. 1a). Subsequent behavioral testing revealed that mice exhibited a significant fear generalization gradient (Fig. 1c), exhibiting maximal immobility when placed in a different chamber scented with MV, less when the chamber was scented with the similar odor butyl acetate (BA, another ester), and still less when the chamber was scented with the dissimilar odor 2-hexanone (HEX, a ketone; one-way repeated-measures ANOVA, $F_{2,44} = 75.82$, $p < 0.001$, $\eta^2 = 0.78$). Control mice placed in these chambers were equally immobile across odors (Fig. 1c), including mice exposed to shocks without odors ($F_{2,24} = 1.83$, $p = 0.18$) and mice exposed to odors without shocks ($F_{2,24} = 0.46$, $p = 0.64$).

We also evaluated relative fear expression across odors in a more naturalistic context by observing the avoidance behavior of mice placed in a square arena with MV, BA, or HEX in each of the corners (Fig. 1b). Control mice did not avoid any of the corners (Fig. 1d, e; one-way repeated-measures ANOVA, $F_{3,33} = 0.53$, $p = 0.66$). By contrast, fear conditioned mice maintained a longer distance from MV and BA, staying closer to the No-Odor corner (Fig. 1d; $F_{3,27} = 30.6$, $p < 0.001$, $\eta^2 = 0.77$) and spending more time in the No-Odor and Hex corners (Fig. 1f). We used information theory-derived metrics to compute the relative likelihood of four a priori models given the avoidance data: no fear of odors, exclusive fear of MV, a similarity-based fear generalization gradient, and overgeneralization of fear equally to all three odors (see Supplemental Methods[46]). The similarity-based fear generalization gradient was the most likely model (AIC 14.2), scoring 7.4-fold more likely than overgeneralization to all odors (AIC 18.2), 11.5-fold more likely than MV-specific fear (AIC 19.1), and 29.1-fold more likely than chance (AIC 21.0). By comparison, in non-conditioned control mice the no fear model (AIC 13.0) was far more likely than any of the competing models (AIC range 19.7–28.3). Taken together these data demonstrate that the fear conditioning paradigm induced fear (Fig. 1c) and avoidance (Fig. 1d–f) of MV that generalized to other odors in proportion to their similarity to MV, a classic sensory generalization gradient[23,47].

If the fear expression and generalization induced by this conditioning paradigm were causally related to a reduction in GABA$_B$ receptor-mediated inhibition in the olfactory bulb, then we would expect to see a) smaller effects of olfactory bulb GABA$_B$ receptor blockade on the response to MV in fear conditioned mice than in odor-only or shock-only control mice, and b) differential effects of GABA$_B$ receptor blockade within individual mice in neuron populations that respond to MV (smallest effects), BA (moderate), and HEX (largest). These changes should manifest throughout the olfactory bulb circuit (Fig. 2c), including its OSN synaptic inputs (where GABA$_B$ receptors are

most prominent[36]) the periglomerular interneurons driven by those inputs, and the mitral cell output of the olfactory bulb. To test these hypotheses, gene-targeted mice conditionally expressing the fluorescent calcium indicator GCaMP6f underwent in vivo optical neurophysiological assessment of odor-evoked neural signaling in populations of OSN terminals, PG cells, and mitral cells of the brain's olfactory bulb (Fig. 2a). Following fear conditioning (or non-conditioned behavioral control) and behavioral testing of fear responses (Fig. 1a), calcium signals evoked by presentation of each odor were measured from the olfactory bulbs of each mouse before and after the sub-dural application of the GABA$_B$ receptor antagonist CGP35348 (or vehicle control) to the exposed dorsal surface of the olfactory bulb. Mice were anesthetized to permit open-skull experimentation.

Local blockade of GABA$_B$ receptors caused an immediate increase in odor-evoked calcium signaling pooled across all mice and all odors (Fig. 2b; one-way ANOVA, $F_{1,1609} = 2860.17$, $p < 0.001$, $\eta^2 = 0.64$)[40]. In control mice that received odor alone- or shock-alone exposures, the effect of GABA$_B$ receptor blockade was nearly identical across odors in OSNs (Fig. 2d, CGP), PG cells (Fig. 2e, CGP), and mitral cells (Fig. 2f, CGP). However, for mice in which MV had been paired with shock the effect of GABA$_B$ receptor blockade was strongly odor-dependent in all three cell types: OSNs (Fig. 2g; odor by training interaction: $F_{2,338} = 20.16$, $p < 0.001$, $\eta^2 = 0.07$; main effect of odor: $F_{2,338} = 11.10$, $p < 0.001$, $\eta^2 = 0.04$), PG cells (Fig. 2h; interaction: $F_{2,540} = 15.49$, $p < 0.001$, $\eta^2 = 0.02$; odor: $F_{2,540} = 51.94$, $p < 0.001$, $\eta^2 = 0.06$), and mitral cells (Fig. 2i; interaction: $F_{2,718} = 19.2$, $p < 0.001$, $\eta^2 = 0.02$; odor: $F_{2,718} = 75.2$, $p < 0.001$, $\eta^2 = 0.09$). The response to the dissimilar odor HEX was comparable between fear conditioned mice and non-conditioned control mice (Fig. 2d–i, black vs threat-predictive odor MV (Fig. 2g–i, purple) and an intermediate effect was observed for the similar odor BA (Fig. 2g–i, blue). These effects are summarized in Fig. 2j, showing a) that odor-cued fear conditioning causes a reduction in GABA$_B$ receptor-mediated inhibition that is greater for odors more similar to the CS and b) this fear learning changes odor-evoked activity across the whole olfactory bulb circuit.

Pooling across mice and odors, we observed a significant negative correlation (Pearson's $r$, $r_{40} = -0.33$, $p < 0.05$) between the size of the physiological effect of GABA$_B$ receptor blockade and the time spent freezing during behavioral testing (Fig. 2k). For non-fear evoking odors in OSNs, the size of the GABA$_B$ receptor blockade effect generally increased over time within a 7-s odor presentation (Fig. 2l, HEX), but this effect was absent for the threat-predictive-odor (Fig. 2l, MV). In all experiments, vehicle application was associated with a small (~5%) reduction in odor-evoked response, likely attributable to response rundown in these post-durotomy experiments.

Generalization of learned responses from one sensory stimulus to another is classically thought to derive from shared elements in the neural representation of similar stimuli[48]. For OSNs and PG cells these elements would be individual glomeruli on the surface of the olfactory bulb, each of which corresponds to a population of OSN inputs expressing a particular odor receptor in the nose[35]. We thus asked whether the fear-associated reduction in GABA$_B$ receptor-mediated inhibition for BA or HEX would be more pronounced in the glomeruli that were also driven by MV. For OSNs, these jointly-responsive glomeruli (which compose the overlapping elements in the glomerular representation of MV with BA or HEX; Fig. 3a), indeed demonstrated significantly less effect of GABA$_B$ receptor blockade in fear conditioned mice than glomeruli that responded only to BA or HEX (Fig. 3c, one-way ANOVA, $F_{1,104} = 9.56$, $p < 0.01$, $\eta^2 = 0.084$). This confirms that some of the neural generalization of the fear-related change in GABA$_B$ signaling from MV to BA is indeed due to overlapping elements in the OSN representation of these odors. However, fear learning also somewhat reduced GABA$_B$ receptor signaling in OSNs in glomeruli that didn't respond to MV at all, with a lesser effect than those that did

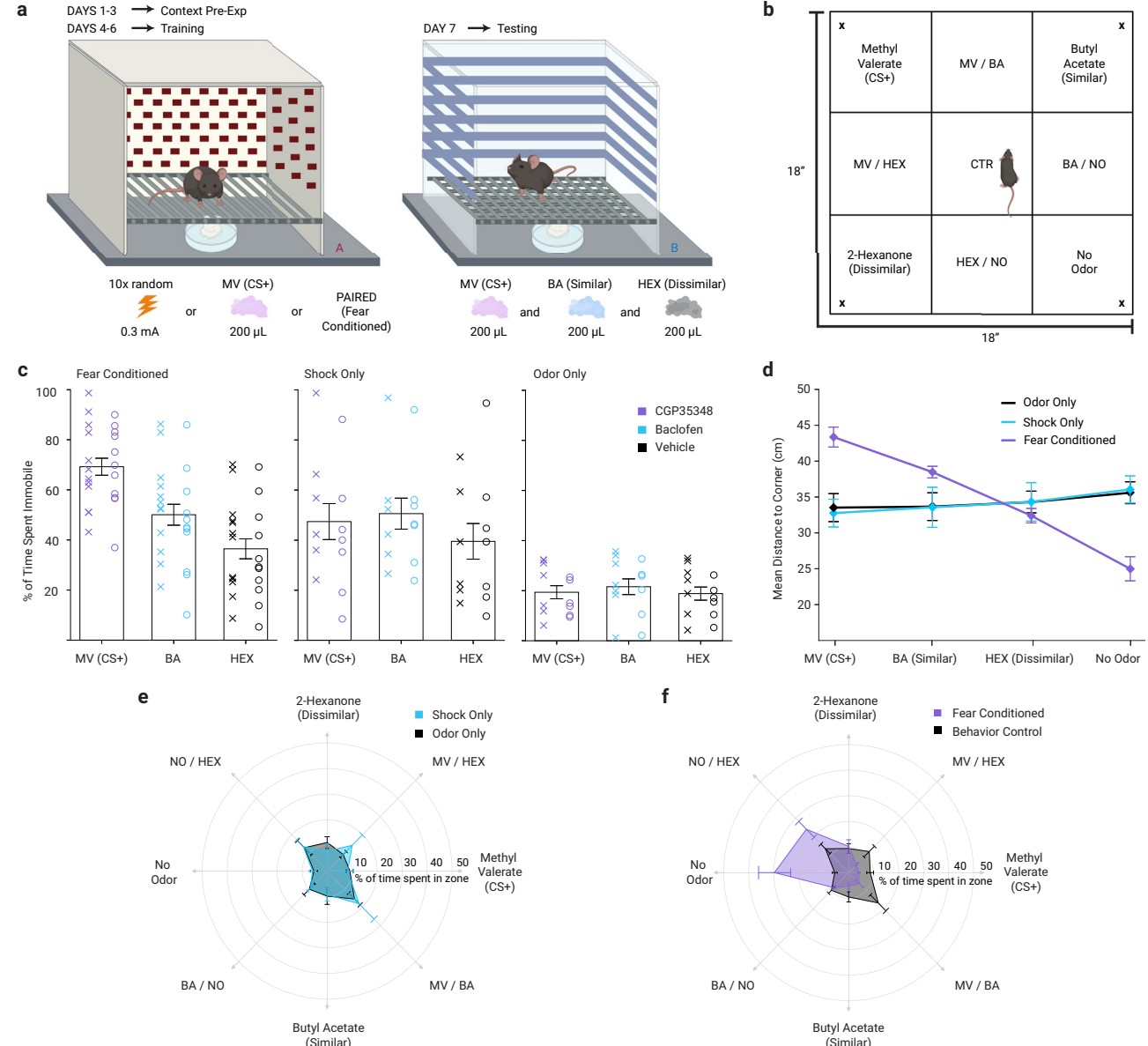

**Fig. 1 | Behavioral setup and baseline behavioral findings. a** Apparatus and experimental design for fear conditioning paradigm and immobility testing. Contextual pre-exposure (first 3 days) and training (next 3 days) were done in context A (shock-only, odor-only or paired) and fear testing was done in context B. Odors were methyl valerate (MV, CS + ), butyl acetate (BA, similar to CS + ), and 2-hexanone (HEX, dissimilar to CS + ). **b** Layout of the open-field arena, locations of odors are marked with "x", equally-sized zones are labeled based on odor position. **c** Bar graphs showing the percentage of time spent immobile in the testing chamber in the presence of each odor, as collected prior to imaging (purple = CGP35348; blue = baclofen; black = vehicle). Data points shown either side of the error bars, x for male and o for female mice. There was no effect of sex on fear gradient in paired animals, (two-way ANOVA, odor x sex interaction, $F_{2,63} = 0.84$, $p = 0.36$, $\eta^2 = 0.08$; two sided). Data are presented as mean ± SEM; $n = 23$ mice (fear conditioned), $n = 13$

mice (shock-only), and $n = 13$ mice (odor-only). **d** Mean distance from each odor corner in the open-field arena for vehicle-infused animals, illustrating preferential avoidance of MV and BA in fear conditioned mice (purple) but not in control groups (blue = shock-only; black = odor-only) **e–f**, Radar plots summarizing the proportion of time spent in each arena zone for vehicle-infused control animals (blue = shock-only; black = odor-only; **e**) and for combined control versus fear conditioned animals (purple = fear conditioned; black = behavioral control; **f**), demonstrating a similarity-dependent spatial avoidance pattern following fear conditioning. Data are presented as mean ± SEM in (**d–f**); $n = 10$ mice (fear-conditioned), $n = 6$ mice (shock-only), and $n = 6$ mice (odor-only). Source data are provided as a Source data file. The schematics in (**a**, **b**) were created in BioRender. Bakir, A. (2026) https://BioRender.com/6u0f62c.

(Fig. 3c, non-overlapping vs non-conditioned controls), suggesting that direct overlap with MV was not the only mechanism of generalization in OSNs. In PG cell populations, the effect of overlapping glomeruli resembled the effect in OSNs (Fig. 3d; one-way ANOVA, $F_{1,174} = 6.41$, $p < 0.05$, $\eta^2 = 0.035$). Mitral cell responses were too spatially diffuse to perform this analysis reliably.

Another potential mechanism contributing to fear generalization across odors could be spatial proximity of the neurons representing

the odors (Fig. 3a), independent of any overlap in the set of neurons responding to each. This was especially of interest in mitral cells, whose secondary dendrites interconnect across long distances in the olfactory bulb. We thus measured the physical distance between each odor-responsive region of interest (ROI) and the ROI that was maximally responsive to MV during imaging of mitral cell populations. As expected, BA-responsive ROIs were closer to the MV maximum than the HEX-responsive ROIs were (Fig. 3b). MV and BA ROIs within 400 μm

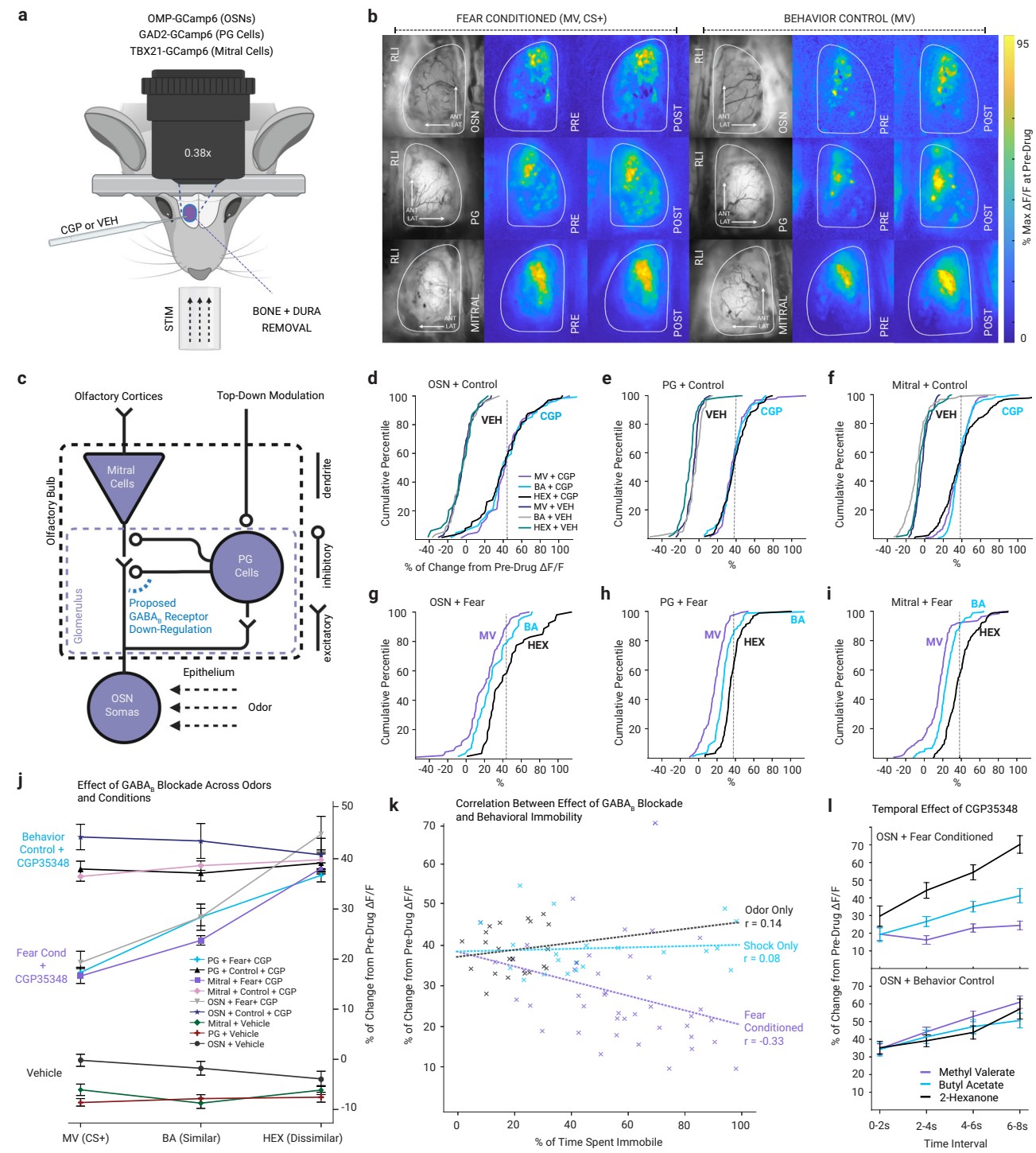

(roughly four glomerular widths) of the MV response maximum showed a smaller increase after GABA$_B$ blockade, in HEX ROI however, this effect was not present (Fig. 3e; two-way ANOVA, interaction $F_{1,699} = 12.00$, $p < 0.001$, $\eta^2 = 0.012$, effect of distance $F_{1,699} = 33.14$, $p < 0.001$, $\eta^2 = 0.033$). This suggests that the effects of fear learning we observed on GABA$_B$-mediated communication were most pronounced near the loci of CS+ activity in mitral cell signaling. Together, these findings suggest that spatial proximity to key regions of neural activity may be playing a significant role in shaping fear generalization through localized changes in GABA$_B$ signaling.

To test whether GABA$_B$ signaling in the olfactory bulb might play a causal role in the expression and generalization of odor-evoked fear,

we implanted bilateral cannulae into the olfactory bulbs and infused GABA$_B$ receptor antagonist (1 mM CGP35348), GABA$_B$ receptor agonist (1 mM baclofen), or vehicle control before testing their odor avoidance behavior. First, we asked whether intrabulbar GABA$_B$ receptor modulation would induce fear or odor avoidance in non-fear conditioned mice. Mice who underwent control training paradigms (odor alone or shock alone) and vehicle control infusions before testing explored the arena freely and spent similar amounts of time in each zone of the arena (Fig. 1e). Mice who underwent the control training paradigms but received intrabulbar infusions of GABA$_B$ receptor antagonist (Fig. 4b) or agonist (Fig. 4c) behaved no differently than those receiving vehicle infusions (Fig. 1e), with no significant differences in overall mobility

**Fig. 2 | Fear learning induces differential effect of GABA$_B$ blockade across odors. a** Optical neurophysiology configuration for widefield calcium imaging of olfactory bulb activity during open-skull experiments. Calcium signals were recorded from olfactory sensory neuron (OSN) axon terminals, periglomerular (PG) cells, or mitral cells using cell-type-specific GCaMP expression while animals received local application of the GABA$_B$ receptor antagonist CGP35348 or vehicle (VEH). Images on the left of each group show baseline fluorescence prior to odor presentation, measured as resting light intensity (RLI). **b** Representative odor-evoked activity maps recorded before (PRE) and after (POST) CGP35348 application in OSNs (top: OPD2 trials 6, 19; OOD1 trials 11, 23), PG cells (middle: GPD3 trials 3,14; GSD1 trials 6, 18), and mitral cells (bottom: MPD5 trials 7, 18; MOD1 trials 10,23) from fear conditioned mice (left) and behavioral control mice (right). Color scale indicates maximum $\Delta F/F$ relative to the pre-drug condition. **c** Schematic of the olfactory bulb circuit highlighting proposed sites of GABA$_B$ receptor–mediated modulation following fear learning. **d**–**i**, Cumulative percentile histograms showing the distributions of effect sizes across all regions of interest and all mice for GABA$_B$ receptor blockade (CGP) or vehicle control (VEH; dark blue = MV; gray = BA; teal = HEX) for each of the three test odors. CGP increased odor response

amplitudes equally for all three odors in OSNs (d), PG cells (e), and mitral cells (f) in non-conditioned control mice but had less effect on MV (the threat-predictive odor) and BA (the similar odor) than HEX (the different odor) in OSNs (**g**), PG cells (**h**) and mitral cells (**i**) from fear conditioned mice (purple = MV; blue = BA; black = HEX). Dashed lines indicate the control median. **j** Composite graph summarizing the effect of GABA$_B$ receptor blockade as a function of odor across all mice. Line colors and symbols indicate cell type and condition as shown in the figure legend; $n = 954$ ROI from 14 mice (behavior control + CGP35348), $n = 1042$ ROI from 14 mice (fear conditioned + CGP35348), and $n = 1229$ ROI from 21 animals (vehicle). **k** Correlations between behavioral and imaging results. Each dot indicates an individual odor/mouse combination. (purple = fear conditioned; blue = shock-only; gray = odor-only). **l** Effect of CGP35348 at different time points within the odor presentation for OSNs from fear conditioned (top) and control (bottom) groups (purple = MV; blue = BA; black = HEX). $n = 191$ ROI (fear conditioned), $n = 207$ ROI (control). Data are presented as mean ± SEM in (**j**, **l**). ROIs were derived from multiple mice; mice represent the biological replicates. Source data are provided as a Source data file. The schematics in (**a**, **c**) were created in BioRender. Bakir, A. (2026) https://BioRender.com/6u0f62c.

(one-way ANOVA, $F_{2, 27} = 3.20$, $p > 0.05$). This demonstrates that GABA$_B$ receptor modulation by itself does not induce fear behaviors like immobility or odor avoidance at these concentrations.

Next, we asked whether intrabulbar inhibitory signaling might influence the expression and generalization of previously learned fear. In control experiments, mice that received MV-cued fear conditioning and vehicle infusions were just as active as non-conditioned control mice ($t_{20} = 0.58$, $p > 0.05$) but exhibited a fear generalization gradient centered on MV as described above (Figs. 1c, d, f and 4d, e). However, when we artificially expanded the fear learning-induced reduction in GABA$_B$ receptor signaling to affect all three odors by infusing the GABA$_B$ receptor antagonist into the olfactory bulb immediately before testing, fear conditioned mice overgeneralized their fear, avoiding all odors. They stayed significantly closer to the unscented corner and farther from the scented corners (Fig. 4f, g), and they avoided HEX just as strongly as MV and BA, spending much more time in the unscented corner and much less time in the HEX corner and the HEX-NO ODOR side of the arena (Fig. 4c–e, h). The best fitting model for odor avoidance in fear conditioned, GABA$_B$ receptor-blocked mice was the overgeneralization model (equal fear of all odors; AIC 11.7), which was 844 times more likely than the generalization gradient model (AIC 25.2) that best described the vehicle control data (Figs. 1 and 4d), while all other competing models were even less likely (AICs 26.7–27.2). This strongly contrasted with the mice who received the GABA$_B$ receptor antagonist after behavioral control treatment (Fig. 4b, gray), with the no fear model (equal time in all zones) fitting the data five orders of magnitude better than any other model (Supplemental Data). This demonstrates that the effect of GABA$_B$ receptor blockade is specific to generalization of fear rather than general odor avoidance.

Finally, we asked whether stimulating GABA$_B$ receptors in the olfactory bulb might reduce fear expression and generalization in olfactory fear conditioned mice by infusing the GABA$_B$ receptor agonist baclofen. We observed generally opposite effects to blocking these receptors using CGP35348 in fear conditioned mice (Fig. 4c). Fear-conditioned, baclofen-infused mice spent somewhat less time in the unscented corner (Fig. 4c–f, h) and more time near BA than vehicle-infused mice. However, they nonetheless exhibited clear avoidance of MV and BA compared to non-conditioned, baclofen-infused mice (Fig. 4c, blue). Their avoidance behavior was not well-described by any of our a priori models (AIC > 22), but the MV-specific fear model and no fear model were both much more likely than either fear generalization model (Supplemental Data). Overall, mice receiving baclofen infusions exhibited less generalized fear than vehicle controls and behaved more similarly to the unconditioned control mice.

Because the effect of intrabulbar infusion of baclofen in fear conditioned mice was to reduce behavioral discrimination among

odors, we performed an additional test to confirm that this dose of baclofen did not impair perceptual discrimination between these odors. Mice were implanted with bilateral intrabulbar cannulae and underwent a habituation/dishabituation test in which mice spontaneously discriminate between a repeatedly presented odor and a novel odor. Each mouse was tested twice, receiving either baclofen or vehicle control infusion prior to testing in randomly assigned order. All mice exhibited decreases in odor investigation across the four habituation trials followed by spontaneous increases in odor investigation upon odor switch (Fig. 5c). Two-way repeated-measures ANOVA revealed a significant main effect of trial sequence as expected ($F_{4,5} = 27.5$, $p < 0.001$; Greenhouse-Geisser corrected), reflecting the habituation/dishabituation sequence (Fig. 5c). There was no significant main effect of drug ($F_{1,5} = 0.82$. $p = 0.408$) or drug x trial-type interaction ($F_{4,5} = 1.022$, $p = 0.420$). These data demonstrate that despite the reduction in olfactory signaling induced by baclofen[38], mice in these behavioral experiments detected and discriminated among the test odors comparably to control animals.

While baclofen had no significant effect overall there was modest, statistically non-significant evidence that baclofen might increase odor habituation relative to vehicle infusion on the fourth odor presentation (Holm-Bonferroni *post-hoc* testing; $df = 20$; $p = 0.110$) prior to a similar or stronger dishabituation on the subsequent odor-switch trial. This suggests that if anything baclofen served to increase odor discrimination.

## Discussion

Taken together, the neurophysiological and behavioral data prove that the downregulation of GABA$_B$ receptor-mediated inhibitory signaling in early olfactory processing is causally linked to the generalization of learned odor-cued fear. The degree of GABA$_B$ downregulation across olfactory bulb glomeruli corresponds to the degree of odor evoked fear (Fig. 2), while the pharmacological suppression of GABA$_B$ receptors in the olfactory bulb induces fear overgeneralization so that mice became afraid of all three odors equally (Fig. 4). The spread of the fear effect to new odors relies on both the overlap in glomerular representations of the similar odor with the threat-predictive odor and the spatial proximity of glomeruli that respond selectively to one odor or another (Fig. 3).

Changes in sensory representations at the input to the brain could have perceptual effects[22,25,27,49]. However, fear learning did not grossly alter the distinctive, odor-specific spatial pattern associated with each odor[40] (Fig. 3a), and mice continued to discriminate robustly among test odors regardless of the infusion of GABA$_B$ receptor agonists (Fig. 5) or antagonists (Fig. 4b). This reaffirms classical evidence that the primary means of generalization is not confusion about what

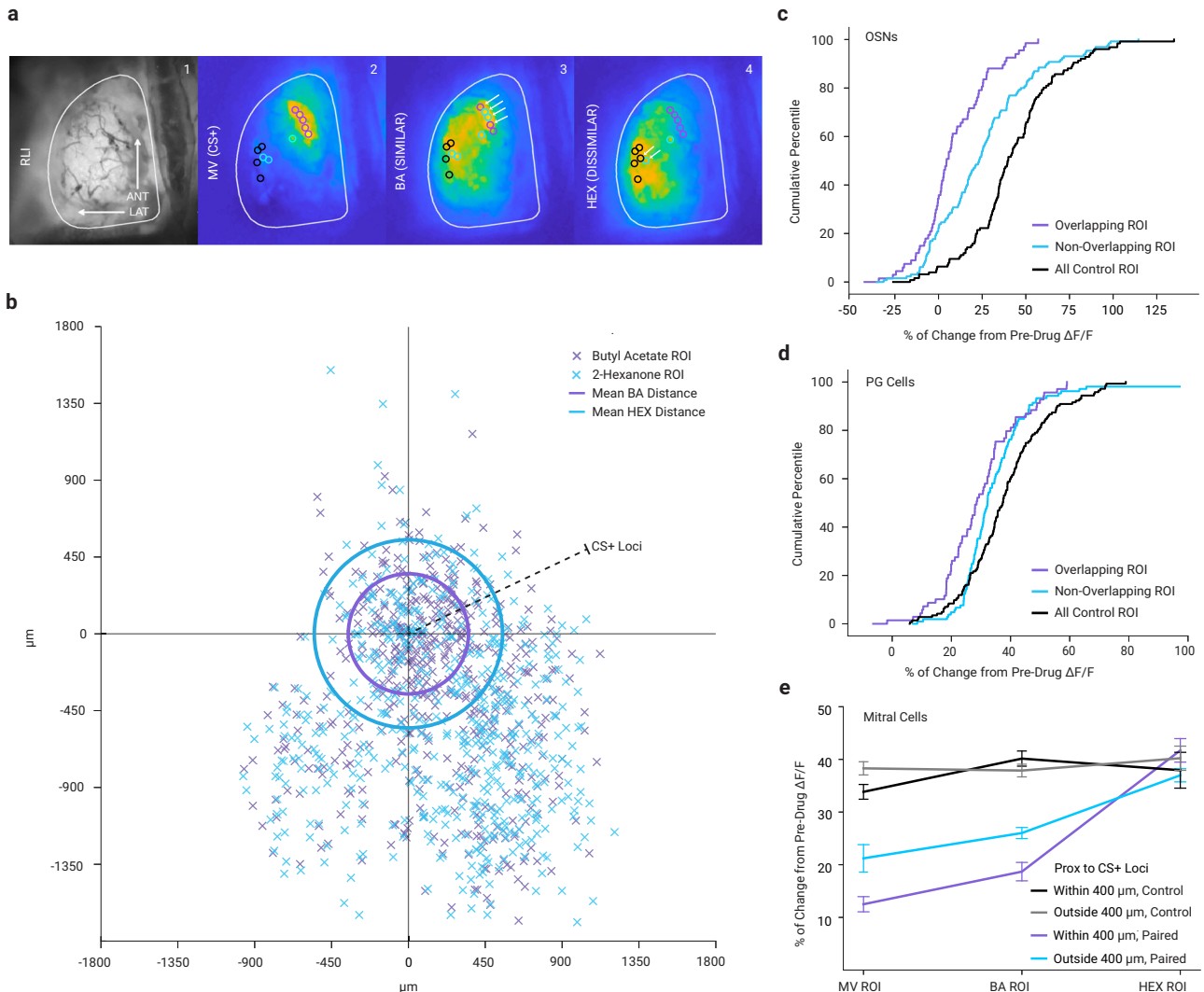

**Fig. 3 | Overlap and distance between glomeruli are important features shaping generalization. a** Example resting fluorescence image (1) and odor-evoked neural activity distributions across the dorsal bulb (MOD1 trials 22, 14, 18) for MV (2) are closer to BA (3) than to HEX (4). Colored circles indicate top 5 responding representative regions of interest (ROIs) identified for each odor (purple = MV, blue = BA, black = HEX). Arrows show jointly-responsive ROI. **b** Anterior-posterior (ordinate) and medial-lateral (abscissa) distances from each non-MV region of interest (ROI, x's) to the maximally responding pre-drug MV ROI in that mouse. Rings indicate mean Euclidian distances from BA to MV (purple) and HEX to MV (blue). For clarity only the most responsive half of glomeruli are displayed. **c–d** Effect of CGP 35348 on ROIs responding to both the threat-predictive odor and another odor (overlapping ROIs, purple) or responding exclusively to another odor (non-overlapping ROIs, blue) compared to non-fear conditioned mice (control, black) in OSN (**c**) or PG populations (**d**). **e** Effect of GABA$_B$ receptor blockade in mitral cells as a function of spatial proximity to the CS$^+$ locus, comparing ROIs located within 400 μm (black = control; purple = paired) or beyond 400 μm (gray = control; blue = paired) of the MV-responsive glomerular region in fear conditioned and control animals. $n$ = 197 ROI from 6 mice (paired + within 400 μm), n = 356 ROI (paired + outside 400 μm), $n$ = 159 ROI (control + within 400 μm), $n$ = 353 ROI (control + outside 400 μm) from 6 mice per group. Data are presented as mean ± SEM in (**e**). ROIs were derived from multiple mice; mice represent the biological replicates. Source data are provided as a source data file.

stimulus is present but changes in the organism's representation of what the present stimulus means[50,51]. Similarly, while generalization is defined behaviorally as an increase in the range of stimuli that can evoke fear or avoidance, there is conceptual ambiguity about whether generalization changes in this assay are better understood as alterations of stimulus-specificity during memory retrieval (i.e. generalization caused by a broadening of the generalization gradient) or as a proportional increase in fear response (i.e. generalization caused by an elevation of the generalization gradient during heightened overall fear). In the avoidance and freezing assays there are floor and ceiling effects that make these mechanisms difficult to disentangle, but alternative designs such as summation tests with compound stimuli (e.g. conditioned inhibition) or competition between behavioral responses (e.g. conditioned suppression of operant behavior) might

illuminate the underlying changes in memory expression induced by olfactory bulb GABA$_B$ receptor modulation[52–54]

In the rodent olfactory bulb, GABA$_B$ receptor expression is very prominent in the terminals of the OSNs[36,55], but there is also expression in PG cells, mitral cells, and external tufted cells[56,57]. Exogenous pharmacological manipulation has effects throughout this inhibitory architecture, so the present methodology does not enable us to attribute fear generalization to plasticity within a single cell population. The most parsimonious explanation of the results may be that the primary drivers of fear learning-induced neuroplasticity are the OSNs, which have direct access to the olfactory stimulus and to peripheral neuroendocrine signals evoked by the aversive shock[58] and which exhibit GABA$_B$ receptor downregulation after odor-cued fear conditioning[31]. This convergence of odor and shock information could

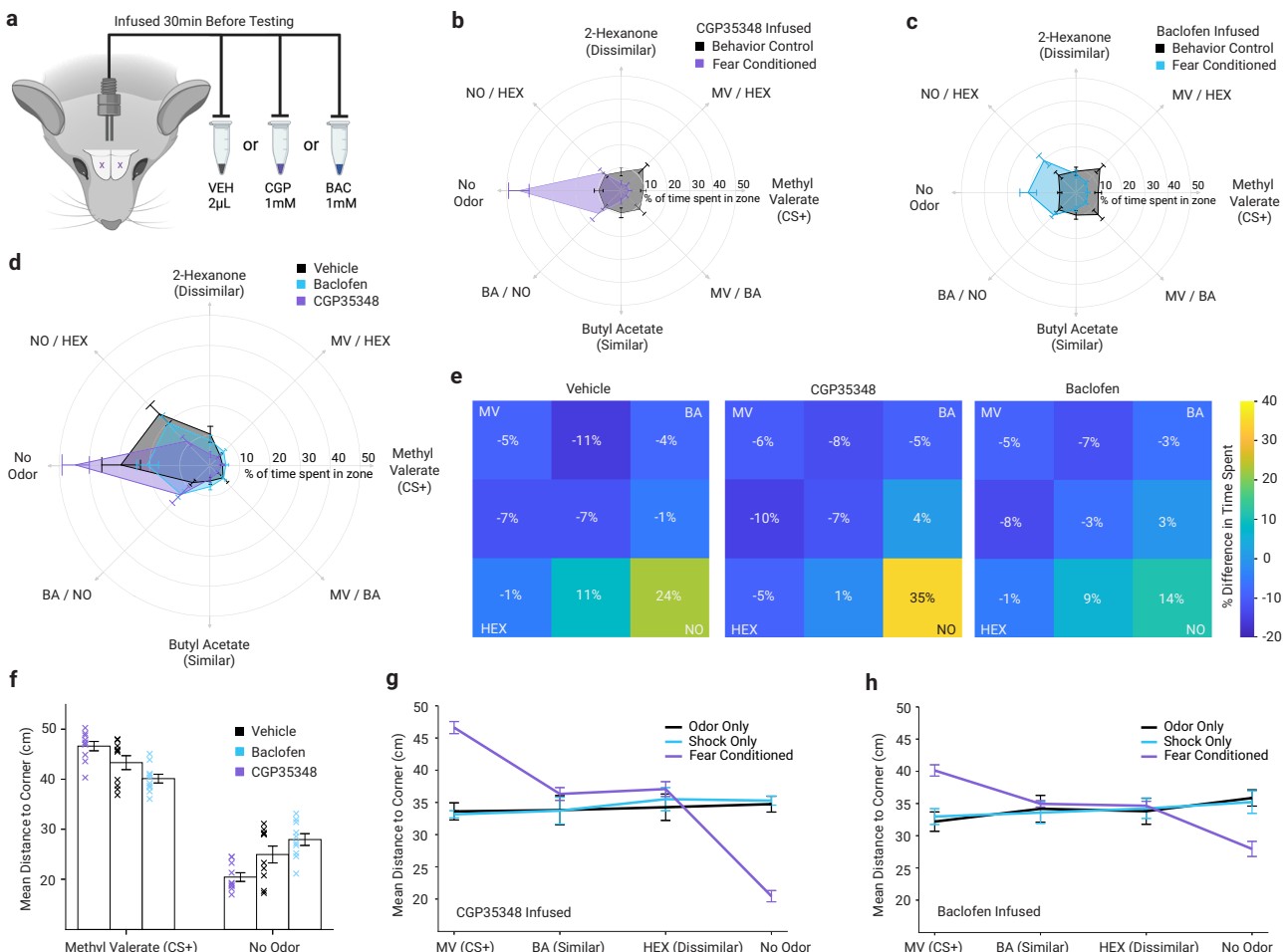

**Fig. 4 | Local GABA_B receptor manipulation in the olfactory bulb shapes fear learning and generalization. a** Schematic illustrating bilateral local infusion of vehicle, the GABA_B receptor antagonist CGP35348, or the GABA_B receptor agonist baclofen into the olfactory bulb 30 min prior to behavioral testing. **b** Radar plot showing the proportion of time spent in each arena zone for CGP35348-infused mice that had previously undergone fear conditioning to methyl valerate (MV; purple) or behavioral control procedures (odor-only and shock-only combined; black). CGP35348 infusion during testing was associated with broadly distributed avoidance across odor locations in fear conditioned mice but not in control animals. **c** Radar plot as in (**b**) for baclofen (GABA_B agonist)-infused mice that received fear conditioned (blue) or control exposures (black), showing reduced avoidance relative to CGP35348-treated animals. **d** Composite radar plot comparing fear conditioned mice infused with vehicle (black; data from Fig. 1), CGP35348 (purple), or baclofen (blue), showing opposing effects of GABA_B receptor blockade and activation on the spatial distribution of avoidance behavior. **e** Heat maps depicting changes in time spent in each arena zone relative to non-conditioned control animals for vehicle, CGP35348, and baclofen-infused mice, illustrating drug-dependent modulation of avoidance patterns. Color scale indicates percent difference in time spent. **f** Mean distance from the conditioned stimulus (MV; CS+) corner and the no-odor corner for fear conditioned animals across infusion conditions (black = vehicle; blue = baclofen; purple = CGP35348). Mean distance to each odor corner for CGP35348-infused (**g**) and baclofen-infused (**h**) animals across fear conditioned (purple), shock-only (blue), and odor-only (black) behavioral groups. Data are presented as mean ± SEM in (**b–d**, **f–h**); n = 22 mice per infusion group (10 fear conditioned, 6 shock-only, 6 odor-only in each). Source data are provided as a source data file. The schematics in (**a**) were created in BioRender. Bakir, A. (2026) https://BioRender.com/6u0f62c.

induce local associative plasticity in the OSNs themselves that in turn drives the fear learning-induced changes observed in downstream neuronal activity (Fig. 2j). Such gating occurring at the level of the GABA_B receptor-expressing sensory afferents could play a privileged role in constraining or amplifying the spread of generalization across similar stimuli. However, the GABA-releasing glomerular circuit and the mitral cells are also highly plastic[32,59–61] and influenced at multiple points by cortical feedback and centrifugal inputs from neuromodulatory structures engaged by fear and other learning tasks[62–64] so the convergence of odor and shock information potentially occurs throughout the olfactory bulb. This would be consistent with the coordinated, odor-dependent effects of both fear learning and GABA_B receptor blockade across OSNs, PG cells, and mitral cells. Regardless of the exact locus of plasticity, these data show that generalization corresponds to changes in how sensory activity is regulated across the early sensory network, with the greatest changes in locations representing the CS+ and diminishing with distance (Fig. 3) across the bulbar circuit.

In the auditory cortex, footshock can evoke a burst of cholinergic neuromodulatory input that excites layer 1 inhibitory interneurons that in turn inhibit a different population of layer 2/3 interneurons, with the ultimate effect of disinhibiting the pyramidal neurons and evoking extra-large responses to sounds accompanied by footshock[45]. The present findings in the olfactory bulb similarly suggest a learning-induced disinhibition of OSNs and mitral cells, within an olfactory bulb circuit that does receive strong centrifugal inputs from other brain regions[28,62,65–68]. The circuit functions demonstrated here are not precisely analogous, in that the olfactory circuit exhibited a persistent, stimulus-specific disinhibition of odor-evoked activity as a mechanism of memory rather than as a response to footshock during conditioning.

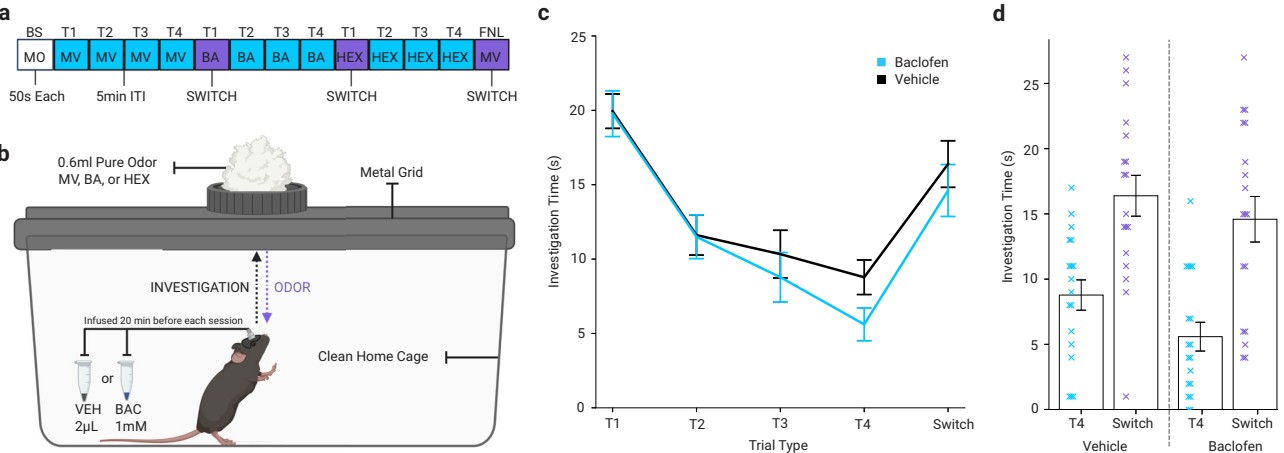

**Fig. 5 | Baclofen infusion does not impair basic odor discrimination in a habituation/dishabituation task. a** Schematic of the habituation/dishabituation trial structure. Following a baseline presentation of mineral oil (MO = white), mice were exposed to four consecutive presentations of a single odor (habituation = blue), followed by four presentations of a second odor and four presentations of a third odor, with a final presentation of the initial odor (50 s per trial; 5 min inter-trial interval). Odors included methyl valerate (MV), butyl acetate (BA), and 2-hexanone (HEX), presented in randomized order across animals. **b** Illustration of the behavioral setup used to measure odor investigation in a clean home cage. Mice received bilateral intrabulbar infusions of baclofen or vehicle 20 min prior to testing, with each mouse tested under both infusion conditions on separate days.

**c** Mean investigation time across repeated presentations of the same odor and following odor switches for vehicle (black) and baclofen-infused (blue) sessions, showing normal habituation across repeated trials and robust dishabituation upon odor change under both conditions. **d** Comparison of investigation time on the final habituation trial (T4 = blue) and the subsequent odor-switch trial (purple) for vehicle- and baclofen-infused sessions, showing stable dishabituation responses in baclofen and vehicle-infused trials. Data are presented as mean ± SEM in (**c**, **d**); *n* = 6 mice, within-subjects design. Source data are provided as a source data file. The schematics in (**b**) were created in BioRender. Bakir, A. (2026) https://BioRender.com/6u0f62c.

However, it may well be that such shock-induced disinhibition is part of the mechanism by odor-shock contingencies selectively alter GABAergic signaling and it is consistent with the idea that local disinhibition can serve the instantiation of associative learning and memory[43] by altering local circuits in precise ways[69]. Ongoing GABA$_B$ receptor-mediated signaling in the olfactory bulb has been linked to avoidance learning during development, even in the absence of explicit odors[70], and the olfactory bulb's interconnections with the amygdala and hippocampal formation could drive aversion or avoidance behavior[71], suggesting that any alterations to the local inhibitory circuitry could have diverse effects on behavior.

GABA$_B$ receptor-mediated presynaptic inhibition is a common architecture across sensory modalities[72–75], so its causal role in shaping fear generalization or overgeneralization may have broad implications for the etiology of generalized anxiety. Anxiety disorders fundamentally reflect overgeneralization of perceived threat to inappropriate stimuli or situations. This can include both explicitly cognitive processes like generalization along conceptual or categorical axes[1,76–78] and explicitly sensory processes like generalization along physical stimulus dimensions[23,79,80], often accompanied by cognitive-sensory sequelae like hypervigilance and hyperresponsiveness to stimuli[21,81]. A key question has been whether the sensory processing changes induced by fear learning are causal to fear generalization or merely reflective of fear generalization[22,82,83]. Here we found that local changes in the magnitude of odor-evoked neurosensory activity as early as the primary sensory neurons and the circuitry of the olfactory bulb can induce behavioral overgeneralization to both similar and dissimilar stimuli. This overgeneralization manifested as pronounced odor avoidance in an entirely different environment than the original traumatic experience, suggesting that the tuning of early sensory representations by GABAergic signaling may have cross-cutting effects on the future behaviors and experiences of the organism. The sensory elements of therapeutic approaches to generalized anxiety may thus prove to be a useful, often less-invasive, target for adjunctive pharmacological or exposure-based treatments.

## Methods

### Mice
The study used a total of 121 adult mice (68 male and 53 female) on a C57BL/6 J background, aged between 3 and 9 months. To image calcium changes, cre recombinase-mediated conditional expression of fluorescent GCaMP6f[84–86] was employed in cells expressing the *Tbet* gene (for mitral cells, a.k.a. *Tbx21*), the *omp* gene (for OSNs), and the *gad2* gene (for PG cells). These transgenic lines were bred and crossed in-house. Non-imaged animals were either from the same batch or wild-type mice of the same C57BL/6 J strain (Jackson Laboratories). All subjects were housed at 21 °C with 40–60% relative humidity under a 12-h light/dark cycle with ad libitum access to food and water. The work was performed in compliance with protocols approved by the Rutgers University Institutional Animal Care and Use Committee (IACUC, protocol #09-022).

### Apparatus
Training cages measuring 10×10 inches (Coulbourn) with shock floors and transparent walls were used, modified for contextual specificity. The open-field setup consisted of a custom-built 18 × 18-inch box with white matte walls and floor, with mesh metal tea strainers suspended in each corner to hold cotton balls with odors. Animals were recorded using a fish-eye lens camera (Opto). Infusions were performed using an infusion tube (Plastics 1) connected to a syringe pump (Harvard Apparatus). Custom cannulas (Plastics 1) included double guide cannulas (1.4 mm), dummy cannulas (protruding 0.15 mm), infusion cannulas (protruding 0.25 mm), and dust caps. Calcium imaging was conducted with a custom-built microscope using an Olympus 4× objective (0.38 NA) and blue LED illumination. Images were captured at 50 Hz with a RedShirtImaging SM-256 CCD camera, providing a resolution of 256 × 256 pixels.

### Experimental design
The study had three different testing conditions, the subjects either underwent calcium imaging (*n* = 49, ages 3–9 months old) with prior

mobility testing, they were tested in the open field ($n = 66$, ages 3–6 months), or they were used in the habituation/dishabituation experiment ($n = 6$, ages 3 to 6 months). The first experiment used a $3 \times 3 \times 2$ design; cell type, training condition, and drug. The training conditions were paired (received both shock and CS + , $n = 23$), shock only (received only shock but not CS + , $n = 13$) and odor only (received only CS+ without any shocks, $N = 13$). The animals have experienced three different odors during the behavioral testing and imaging sections; methyl valerate (CS + ), butyl acetate (a similar odor to CS + , both fruity-smelling ester), and 2-hexanone (quite different from the CS + , a chemical-smelling ketone). The CS+ odor was methyl valerate in all training conditions. Subjects were also divided into two drug condition groups according to what type of solution was applied on top of the olfactory bulb halfway through the imaging sessions; 28 of the subjects received 1 mM CGP 35348 dissolved in Ringer's solution while the remaining ($n = 21$) received vehicle (Ringer's solution). The open-field subjects ($n = 66$) also underwent the same training paradigm, but they received CGP 35348 ($n = 22$), baclofen ($n = 22$) or vehicle (Ringer's, $n = 22$) infusions into their olfactory bulbs prior to testing. The experimenter was blind to the contents of all infusates.

### Training and testing

Fear learning was established using two visually distinct contexts. Context A (training) featured red dots on opaque white glass, a vertically lined shock floor, and a red drop tray. Context B (used only for calcium imaging) had transparent glass walls, blue stripes, visually different floors, and a gray drop tray. The animals who received the training were kept at separate housing shelves or had other cages in between, this was done to prevent any external learning from happening because of the visible stress of animals nearby. For the first three days, mice were placed in Context A without any stimuli for context pre-exposure. On the next three days, they were again placed in Context A, but this time with their group-specific stimuli. During training, the CS+ odor (methyl valerate) was always present in the context. Two hundred microliters of methyl valerate were applied to a cotton ball with a half-inch diameter, which was placed in a small weighing boat positioned between the shock floor and the drop tray. (During context pre-exposure, the weighing boat and cotton ball were placed in the same position without any odor). Mice in the US groups (paired and shock-only) received 10 mild shocks (0.3 mA) at random intervals (40-80 seconds). On the seventh day, calcium imaging animals were tested in Context B. Testing sessions included three sets of 120 s, each with a different odor: methyl valerate, butyl acetate, or 2-hexanone. The order of odor presentations was randomized via a code to avoid matching the imaging order. The first 30 s of each session were excluded from analysis, assuming the animals were still adjusting to the environment. During the remaining 90 seconds, their movements were recorded to calculate the percentage of immobile time. Open-field animals were tested after drug infusions (see Surgery and Infusion Parameters). Within the open-field box, a different odor was placed in each of the four corners. As in training, a cotton ball treated with 200 microliters of pure odor was placed inside a $2 \times 2$-inch metal tea bag hung in each corner. The odors included methyl valerate, butyl acetate, 2-hexanone, and no odor. Methyl valerate and butyl acetate were always placed in adjacent corners to create a "danger zone," while 2-hexanone and the no-odor corner formed a "safe zone." Animals were free to explore the box for 5 min, after which they were returned to their individual cages.

### Surgery and infusion parameters

In the calcium imaging group, mice underwent surgery on the same day, a few hours after behavioral testing, to minimize the impact of the testing experience on plasticity. Mice were anesthetized using pentobarbital (10 mg/kg) administered intraperitoneally, and their scalps were shaved in preparation for surgery. Body temperature was maintained at $38 \pm 0.5\,°C$ throughout the procedure. To reduce mucus secretion and brain swelling, atropine (100 mg/kg) and dexamethasone (3 mg/kg) were administered, respectively. Additionally, 0.25% bupivacaine (0.25 mL) was applied as a local anesthetic before scalp removal. The periosteal membrane was scraped off, and 70% ethyl alcohol was applied to dry the skull. To ensure stability, the thickest part of the skull was drilled slightly using a coarse dental bit to create a shallow burrow for the adhesive. The mouse's head was fixed in place using a head bar, super glue, and dental acrylic. A well-like structure was built around the olfactory bulb using dental acrylic to hold the vehicle or drug in place. Before craniotomy, the bone over the olfactory bulb (both sides) was thinned with a coarse dental bit. One side of the bulb remained intact as a control. For the drilled side, a cooled Ringer's solution was used during the operation to prevent overheating and to avoid air contact with the dura mater. A piece of bone covering approximately 70%-80% of the area was removed using an inlay prep burr. After the dura mater was exposed, a microscopic incision was made using a handmade dura hook (a slightly bent 30-gauge needle), and the dura was carefully removed with fine-tipped forceps. Any bleeding was controlled with surgical foam, and cleaning was kept minimal to avoid damage to the exposed brain. A glass coverslip was placed over the exposed area, floating on the Ringer's solution, before imaging.

For the open-field and habituation/dishabituation groups, cannula insertion surgery was performed before training[87]. The procedure followed the same initial steps as described above. Two reference points were marked parallel to the center of the olfactory bulb, where drilling was performed. Guide cannulas were inserted into these drilled holes along with dummy cannulas and dust caps. The exposed skull was encased in acrylic solution. Animals were given 1 mL saline and 10 mg/mL carprofen for recovery and allowed 2–4 days of rest before training. Before testing, animals were anesthetized with isoflurane, and infusion tubes were inserted into their guide cannulas to deliver CGP 35348, baclofen, or vehicle (Ringer's solution) for open-field animals and baclofen or vehicle for habituation/dishabituation animals. For vehicle-treated animals, 2 μL of Ringer's solution was infused over 1 min. For experimental animals, 2 μL of either 1 mM CGP 35348 or 1 mM baclofen in Ringer's solution was infused using the same protocol. After infusions, animals rested for 20 min before testing.

### Habituation/dishabituation behavioral test

To evaluate odor discrimination in baclofen-infused mice, we used a habituation/dishabituation paradigm, designed based on a previously established protocol[49]. Each mouse ($n = 6$) was tested under both baclofen and vehicle infusion conditions on separate days, with 2–6-day interval between tests. On each test day, mice received an infusion of either baclofen or vehicle, followed by a 20-min rest period before testing. The experimenter remained blind to the solution type used for infusion throughout all procedures. Mice were tested in a clean home cage measuring $5 \times 7 \times 11.5$ inches and were video recorded using a fisheye lens camera (Opto) positioned 5 inches over the cage. Each session began with a single presentation of a blank tea strainer containing a cotton ball treated with 0.6 mL mineral oil to habituate the mouse to the stimulus presentation method. This was followed by thirteen 50 s presentations of 0.6 mL pure odorant delivered on a cotton ball placed in a mesh metal tea strainer manually positioned in the center of the wire cage top. Each trial was separated by 5 min. The trial sequence included 4 trials each of three test odors (MV, BA, and HEX), and a final trial of the 1st odor, yielding 6 possible odor sequences randomly assigned across mice. Video recordings were manually scored by an experimenter blind to the experimental condition. Investigation time on each trial was defined as time spent in a rearing, nose-upward posture anywhere in the cage.

## Optical imaging

The optical imaging preparation[30,88] was performed as follows. After surgery, subjects were placed under the camera setup mentioned earlier. Odor delivery was controlled through an odor tube positioned directly in front of the animal's nose. A MATLAB (MathWorks, R2019a) code managed the odor delivery system, which provided nitrogen (odor carrier), ultra-zero air (dilution), and the target odor. The odor concentration and dilution parameters were determined by computer-controlled mass-flow controllers. The first step of each imaging session was to compare the odor-evoked signals on the two sides of the olfactory bulb (thinned vs. durotomy side). If an imbalance was observed due to surgical complexity, the animal was returned to the surgery area, and the same surgical procedure was performed on the intact side. Minor to moderate damage or bleeding that did not affect signaling was overlooked. Where necessary, the exposed area was washed with Ringer's solution between trial blocks to avoid bleeding. Each trial consisted of 4.5 s of pre-odor baseline, 7 seconds of odor stimulation, and 8.5 s of post-odor period, resulting in a total trial duration of 20 seconds (1000 frames at 50 Hz: 225 pre-odor, 350 odor, 425 post-odor). Each trial block contained 8 trials: the first 4 were mock odor presentations at 2% strength to minimize novelty effects, while the last 4 were recorded for analysis. Animals were kept under anesthesia with boosters administered as needed. To avoid interference with the imaging process and breathing, the boosters were given in smaller amounts (~0.01–0.02 mL vs. ~0.05 mL) but at more frequent intervals (~15 min vs. ~30 min). Boosters were administered before mock trials, and these trials were excluded from analysis. The imaging sessions involved the same three odors used during behavioral testing: methyl valerate (CS+), butyl acetate (similar), and 2-hexanone (different). Six trial blocks were performed for each animal: three during the baseline period and three following drug application. Trials were separated by 60–90 s inter-trial intervals. After three baseline trial blocks, the Ringer's solution on the exposed bulb was removed with a transfer pipette and replaced with either CGP 35348 (1 mM in Ringer's) or vehicle (Ringer's). The same trial structure was repeated after drug application. The experimenter conducting the imaging was blind to the drug condition.

## Data analysis and visualization

Imaging data were analyzed using custom-written codes[30,89] with additional scripts for spatial pattern analysis. Folder and filenames were automatically shuffled and randomized to ensure the experimenter was blind to the experimental conditions during analysis. Data processing was performed using MATLAB (R2019a) and Python (v3.8). Odor-evoked fluorescence responses ($\Delta F$) were calculated by subtracting the mean baseline fluorescence (signal within 1 second before the start of odor presentation) from the fluorescence level in each frame. The normalized value ($\Delta F/F$) was obtained by dividing $\Delta F$ by the baseline fluorescence. Before analyzing individual trials, glomerular regions of interest (ROIs) were manually selected using average maps from each animal. For mitral cells, regions with shapes resembling glomeruli were selected because their deeper location resulted in fuzzier signals. Each animal and odor combination had its own unique set of ROIs. For each trial and ROI, peaks in odor-evoked signals and their spatiotemporal specifics were identified and averaged. The main dependent variable was the percentage increase in $\Delta F/F$ from pre-drug to post-drug for each ROI. For spatial analyses, the location of each ROI was calculated to evaluate overlap and proximity. Overlap was defined as CS- ROIs located within 70 μm (average size of a small glomerulus) of CS+ ROIs in each animal. Proximity was determined by calculating the Euclidean distance of each CS- ROI from the max $\Delta F/F$ CS + ROI for that animal. ROIs were further categorized based on whether they were within 400 μm (approximately the diameter of four glomeruli) of the CS + ROI or farther away. For temporal analyses, the post-odor presentation fluorescence signal of each ROI was divided into four

2-second intervals. The maximum $\Delta F/F$ value within each interval was calculated. Behavioral data were analyzed using EthoVision software (XT 14). Pre-imaging mobility analysis was conducted to calculate the percentage of immobile time for each animal. To confirm fear learning and generalization, paired animals that did not exhibit specific aversion were excluded from imaging. The exclusion criterion required at least a 5% immobility difference between the CS+ odor and other odors. These percentages were then used to assess fear learning and generalization, as well as to correlate behavioral findings with imaging data. For open-field animals, two main analyses were conducted: time spent in different zones and the average distance from specific corners. The open-field arena was divided into 9 equal square zones, including odor zones, intermediate zones, and the center. The time spent in each zone was determined based on the animal's center point in the video. All statistical analyses were performed in RStudio (v2024.02.0), and graphs were created using Python (v3.8) and Adobe Illustrator (2025). Illustrations were drawn using bioRender and Adobe Photoshop (2025).

## Reporting summary

Additional details about the research design can be found in the Nature Portfolio Reporting Summary associated with this article.

## Reporting summary

Further information on research design is available in the Nature Portfolio Reporting Summary linked to this article.

## Data availability

Source data underlying the figures are provided with this paper. Processed datasets and analysis code are available at Zenodo (https://doi.org/10.5281/zenodo.18809357). Raw imaging data are not publicly available due to their large file size and will be made available by the corresponding author upon request. Source data are provided with this paper.

## Code availability

Code used for analyzing the data is deposited at Zenodo (https://doi.org/10.5281/zenodo.18809357).

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

## Acknowledgements

The authors would like to thank David J. Barker, Paul A.S. Breslin, and Louis D. Matzel for their comments on earlier versions of this work. This work was supported by the National Institute of Mental Health and National Institute on Deafness and Other Communication Disorders (R01MH101293, J.P.M.).

## Author contributions

A.K.B. and J.P.M. conceptualized the study, designed the experiments and wrote the manuscript. A.K.B. performed the surgeries, experiments, and analyses. A.A.G., N.E., and K.L.L.C. assisted with cannula surgeries, analysis, training, and testing in the open-field arena. M.C.R., A.A.G., and K.L.L.C. contributed to pilot work and development of the open-field setup.

## Competing interests

The authors declare no competing interests.
