## [Transparent Peer Review file · Nature Communications]

Generalization of fear learning is shaped by inhibitory sensory processing in mice

Corresponding Author: Dr Alper Bakir

Version 0:

Reviewer comments:

Reviewer #1

(Remarks to the Author)

This study by Bakir et al. investigates the role of the early olfactory system in fear learning by examining activity in olfactory sensory neurons (OSNs), mitral cells (MTs), and periglomerular (PG) cells within the olfactory bulb (OB). By manipulating GABA receptor activity through the use of an antagonists and an agonist, the authors demonstrate that inhibitory processing in the early olfactory pathway plays a critical role in fear learning and, potentially, early memory formation. This finding significantly extends previous work on sensory disinhibition—particularly in the primary auditory cortex—by demonstrating a similar mechanism in early olfactory areas. The study is methodologically rigorous, and the results largely support the authors' conclusions. However, I offer several suggestions for further strengthening the manuscript:

1. Conceptual framing: To broaden the theoretical context, the authors could integrate earlier work on disinhibition in the auditory cortex. Specifically, the following studies may be useful in anchoring the current findings within a broader framework for associative fear learning and memory:
 - o Letzkus, J. J., Wolff, S. B., & Lüthi, A. (2015). Disinhibition, a circuit mechanism for associative learning and memory. *Neuron*, 88(2), 264–276.
 - o Letzkus, J. J., Wolff, S. B., Meyer, E. M., et al. (2011). A disinhibitory microcircuit for associative fear learning in the auditory cortex. *Nature*, 480(7377), 331–335.
2. Result interpretation: The pharmacological manipulation in the olfactory bulb affects multiple cell types (OSNs, PG cells, and MT cells), making it difficult to isolate the causal role of OSNs in fear learning. While the title appropriately avoids specifying OSNs, the Introduction places almost exclusive emphasis on them, and the Discussion does not sufficiently explore alternative explanations involving PG or MT cell contributions. A more balanced consideration of these possibilities would strengthen the overall interpretation.
3. Fear generalization: While the generalization gradient observed is compelling, the data do not conclusively rule out the alternative interpretation that disinhibition directly influences fear learning and only secondarily affects generalization. This distinction could be more clearly addressed in the discussion.
4. Figure presentation (minor point): Some figure captions would benefit from additional explanation to help readers interpret the results more effectively without referring back to the main text.

(Remarks on code availability)

Reviewer #2

(Remarks to the Author)

Bakir et al. investigated the neural mechanism underlying fear generalization using an odor (CS+)-mediated, contextual fear conditioning paradigm in mice. The authors show that odor fear conditioning induced fear responses (immobility) and odor avoidance in an open field arena, which can be generalized to other odors based on their similarity to the CS+ odor. This generalization is linked to overlapping and spatial proximity within the olfactory bulb activation map. The authors further show that fear generalization is regulated by GABAB receptor signaling in the olfactory bulb via pharmacological manipulations. Blocking GABAB receptors with an antagonist in fear-conditioned mice leads to broader fear generalization to other odors, while infusion of a GABAB agonist shows reduced fear generalization. The authors demonstrate that

downregulation of GABAB receptor-mediated inhibition enhances fear generalization via representational plasticity rather than sensory confusion. The overlap and spatial relationship between glomerular responses to different odors underpin the spread of fear. Given the conserved role of GABAB signaling across sensory systems, these results suggest a broader relevance for understanding mechanisms underlying generalized anxiety disorders. The study is well-designed with appropriate controls and statistical analyses and is clearly written. The manuscript would be further improved if the following concerns were addressed.

Major:

1. Activation of GABAB receptors in the olfactory bulb with baclofen is expected to reduce glutamate release from OSNs. Therefore, baclofen-induced behavioral effects shown in Fig. 4 could be potentially due to altered odor detection/discrimination, but not failed fear generalization to other odors. Olfactory bulb imaging experiments (parallel to those shown in Fig. 2B-I for the GABAB antagonist) and/or behavioral tests to show that mice can still distinguish the three odors would help to rule out this possibility and further strengthen the conclusions.
2. In addition to the OSN axon terminals, GABAB receptors are also expressed in other cell types in the olfactory bulb. The authors should discuss potential contributions from other cell types to fear generalization after exogenous application of GABAB agonists and antagonists.

Minor:

1. Both male and female mice are used in the study. Is there a sex difference observed in the mouse behavior? It would be helpful to use different symbols for male and female mice in Fig. 1C. For example, use x for males and circles for females.
2. In all figure legends, the animal numbers and trial numbers should be provided wherever applicable.
3. In reporting one-way ANOVA results, n_2 or η^2 is used in different places to represent eta-squared. Please make it consistent throughout.
4. In Figure 2 or Figure 3, it would be helpful to show examples of ROIs. This will help readers to appreciate how much overlap there is across the activation patterns induced by different odors.
5. In Fig. 2C, suggest using "OSN Somas" instead of "OSN Bodies".
6. In Fig. 2B, in the label "BEHAVIOR CONTROL (CS+)", there is no CS+ odor in the control groups. Suggest removing "CS+" but using MV (the odor name instead). Please define RLI in the figure legend.
7. Page 5. Last sentence. "odor-evoked responding", suggest changing to "odor-evoked response".
8. Description of Fig. 3A legend seems inaccurate. It states "...across the dorsal bulb for MV (2nd row) are closer to BA (3rd row) than to HEX (bottom)", while all three images are in the same row.
9. Page 11. "...OSN inputs to the bub", where "bub" should be "bulb".
10. On page 7, "... the effect of overlapping glomeruli resembled the effect in OSNs (Fig. 2D, ...". It should be Fig. 3D instead of Fig. 2D.
11. In Fig. 4E legend, "with color scale bar a right" should be "...at right".

(Remarks on code availability)

Reviewer #3

(Remarks to the Author)

(Remarks on code availability)

Version 1:

Reviewer comments:

Reviewer #1

(Remarks to the Author)

The authors have adequately addressed my concerns.

(Remarks on code availability)

Reviewer #2

(Remarks to the Author)

The authors have satisfactorily addressed the critiques raised in the initial review. We have no further comments.

(Remarks on code availability)

Reviewer #3

(Remarks to the Author)

(Remarks on code availability)

RESPONSE TO REVIEWERS

We are grateful to the reviewers and the Editor for their kind words regarding the quality and impact of this work, and we thank them for their helpful comments. We have made targeted revisions throughout the text, including the incorporation of data from a new control experiment and the addition of new discussion to incorporate reviewer suggestions regarding cell type-specific contributions and disinhibitory circuit mechanisms for learning in other systems. We have also made clarifying updates to figures and captions. These changes are detailed below.

Reviewer #1:

This study by Bakir et al. investigates the role of the early olfactory system in fear learning by examining activity in olfactory sensory neurons (OSNs), mitral cells (MTs), and periglomerular (PG) cells within the olfactory bulb (OB). By manipulating GABA receptor activity through the use of an antagonists and an agonist, the authors demonstrate that inhibitory processing in the early olfactory pathway plays a critical role in fear learning and, potentially, early memory formation. This finding significantly extends previous work on sensory disinhibition—particularly in the primary auditory cortex—by demonstrating a similar mechanism in early olfactory areas. The study is methodologically rigorous, and the results largely support the authors' conclusions. However, I offer several suggestions for further strengthening the manuscript:

1. Conceptual framing: To broaden the theoretical context, the authors could integrate earlier work on disinhibition in the auditory cortex. Specifically, the following studies may be useful in anchoring the current findings within a broader framework for associative fear learning and memory:

*o Letzkus, J. J., Wolff, S. B., & Lüthi, A. (2015). Disinhibition, a circuit mechanism for associative learning and memory. *Neuron*, 88(2), 264–276.*

*o Letzkus, J. J., Wolff, S. B., Meyer, E. M., et al. (2011). A disinhibitory microcircuit for associative fear learning in the auditory cortex. *Nature*, 480(7377), 331–335.*

We thank the reviewer for their kind words about our methodological rigor and the strength of our conclusions. We also appreciate their thoughtful synthesis of our findings within the broader literature on sensory disinhibition and associative learning. We now discuss disinhibition of local circuitry during learning as a recurring circuit motif in the Introduction, including direct reference to the auditory cortex as suggested (page 2, paragraph 5). We have also added a full paragraph to the discussion exploring the similarity of our findings to those reported in the auditory system (page 13, paragraph 3).

2. Result interpretation: The pharmacological manipulation in the olfactory bulb affects multiple

cell types (OSNs, PG cells, and MT cells), making it difficult to isolate the causal role of OSNs in fear learning. While the title appropriately avoids specifying OSNs, the Introduction places almost exclusive emphasis on them, and the Discussion does not sufficiently explore alternative explanations involving PG or MT cell contributions. A more balanced consideration of these possibilities would strengthen the overall interpretation.

We of course agree that pharmacological manipulation of GABA_B receptors in the olfactory bulb affects multiple cell types and we thank the reviewer for bringing this oversight to our attention. We have added a sentence to the Introduction to establish this possibility up front (page 2, paragraph 4) and a full paragraph to the Discussion (page 13, paragraph 2) to explicitly consider our data in light of these broader effects on the circuit.

3. Fear generalization: While the generalization gradient observed is compelling, the data do not conclusively rule out the alternative interpretation that disinhibition directly influences fear learning and only secondarily affects generalization. This distinction could be more clearly addressed in the discussion.

The pharmacological manipulations reported here were performed only after conditioning had already occurred, but the reviewer is correct that the interpretation of the behavioral generalization we observed could differ depending on one's underlying conceptual understanding of generalization itself. We have added a full paragraph to the discussion exploring the disparate possibilities of generalization linked to stimulus confusion, generalization linked to gradient broadening, and generalization linked to gradient elevation (page 12, paragraph 2) and suggesting potential means of disentangling these possibilities.

4. Figure presentation (minor point): Some figure captions would benefit from additional explanation to help readers interpret the results more effectively without referring back to the main text.

We thank the reviewer for this suggestion and have revised the figure captions throughout the manuscript to provide additional explanatory detail, clarify experimental context, and improve stand-alone interpretability without requiring reference to the main text.

Reviewer #2:

Bakir et al. investigated the neural mechanism underlying fear generalization using an odor (CS+)-mediated, contextual fear conditioning paradigm in mice. The authors show that odor fear conditioning induced fear responses (immobility) and odor avoidance in an open field arena, which can be generalized to other odors based on their similarity to the CS+ odor. This

generalization is linked to overlapping and spatial proximity within the olfactory bulb activation map. The authors further show that fear generalization is regulated by GABAB receptor signaling in the olfactory bulb via pharmacological manipulations. Blocking GABAB receptors with an antagonist in fear-conditioned mice leads to broader fear generalization to other odors, while infusion of a GABAB agonist shows reduced fear generalization. The authors demonstrate that downregulation of GABAB receptor-mediated inhibition enhances fear generalization via representational plasticity rather than sensory confusion. The overlap and spatial relationship between glomerular responses to different odors underpin the spread of fear. Given the conserved role of GABAB signaling across sensory systems, these results suggest a broader relevance for understanding mechanisms underlying generalized anxiety disorders. The study is well-designed with appropriate controls and statistical analyses and is clearly written. The manuscript would be further improved if the following concerns were addressed.

We thank the reviewer for their praise of our experimental design and sincerely appreciate their suggestions, which we have incorporated throughout.

Major:

1. Activation of GABAB receptors in the olfactory bulb with baclofen is expected to reduce glutamate release from OSNs. Therefore, baclofen-induced behavioral effects shown in Fig. 4 could be potentially due to altered odor detection/discrimination, but not failed fear generalization to other odors. Olfactory bulb imaging experiments (parallel to those shown in Fig. 2B-I for the GABAB antagonist) and/or behavioral tests to show that mice can still distinguish the three odors would help to rule out this possibility and further strengthen the conclusions.

The reviewer makes an excellent point, and we have performed an additional behavioral experiment (new Fig. 5) to address this concern. We used an identical infusion of baclofen into the olfactory bulbs of mice prior to passive presentations of the three study odorants using a within-subjects habituation/dishabituation paradigm (Methods: page 16, paragraph 2 and Results: page 11, paragraph 4). Baclofen did not impair odor discrimination at all: mice that received baclofen infusions exhibited normal habituation to repeated odor presentations and robust dishabituation upon odor change, with no significant main effect of drug vs vehicle control or drug-by-trial interaction during odor switches. This confirms that these mice could detect and discriminate these odors and supports the interpretation that its effects on behavior reflect modulation of fear generalization rather than sensory confusion. We believe these data are sufficient to address the reviewer's concern. We have not added physiological experiments with baclofen because a) physiological effects of baclofen on this circuit have been reported dozens of

times previously by ourselves and others in multiple labs and species (e.g. Wachowiak et al. 2005; McGann et al. 2005), and b) the effects of exogenous agonists are much harder to interpret than the effects of antagonists blocking naturally occurring signaling.

J. P. McGann et al., Odorant representations are modulated by intra- but not interglomerular presynaptic inhibition of olfactory sensory neurons. *Neuron* 48, 1039-1053 (2005).

M. Wachowiak et al., Inhibition of olfactory receptor neuron input to olfactory bulb glomeruli mediated by suppression of presynaptic calcium influx. *Journal of neurophysiology* 94, 2700-2712 (2005).

2. In addition to the OSN axon terminals, GABAB receptors are also expressed in other cell types in the olfactory bulb. The authors should discuss potential contributions from other cell types to fear generalization after exogenous application of GABAB agonists and antagonists.

We thank the reviewer for bringing this oversight to our attention and have added text to both the Introduction and Discussion as described in the response to Reviewer 1.

Minor:

1. Both male and female mice are used in the study. Is there a sex difference observed in the mouse behavior? It would be helpful to use different symbols for male and female mice in Fig. 1C. For example, use x for males and circles for females.

We thank the reviewer for reminding us to include these analyses where possible. We did not observe any sex differences in the pre-imaging behavioral experiments. We have updated Fig. 1c to use different symbols for male and female mice and added the corresponding statistical information to the figure caption. The optical neurophysiology data from these mice similarly showed no detectable sex differences, $F_{2,63} = 0.22$, $p = 0.80$, $\eta^2 = 0.006$. For the highly multivariate open-field avoidance experiments, our sample sizes were insufficient to support *post hoc* analyses of potential sex differences, so these are not reported in accordance with the journal's guidelines for consideration of sex as a biological variable.

2. In all figure legends, the animal numbers and trial numbers should be provided wherever applicable.

These numbers have been added throughout.

3. In reporting one-way ANOVA results, n^2 or η^2 is used in different places to represent eta-squared. Please make it consistent throughout.

Thank you for pointing out this typographical error. We have corrected the manuscript to report eta-squared consistently as η^2 for all one-way ANOVA results.

4. In Figure 2 or Figure 3, it would be helpful to show examples of ROIs. This will help readers to appreciate how much overlap there is across the activation patterns induced by different odors.

We have revised Fig. 3a to include representative examples of odor-responsive ROIs, showing the top five maximally responding ROIs for each odor and marking examples of jointly responsive glomeruli with arrows, to better illustrate overlap across odor-evoked activation patterns.

5. In Fig. 2C, suggest using “OSN Somas” instead of “OSN Bodies”.

We have revised the label in Fig. 2c to read “Somas.”

6. In Fig. 2B, in the label “BEHAVIOR CONTROL (CS+)”, there is no CS+ odor in the control groups. Suggest removing “CS+” but using MV (the odor name instead). Please define RLI in the figure legend.

We have revised the label in Fig. 2B to remove “CS+” and instead use the odor name (MV), and we have added a definition of RLI to the figure legend.

7. Page 5. Last sentence. “odor-evoked responding”, suggest changing to “odor-evoked response”.

We have corrected this phrasing.

8. Description of Fig. 3A legend seems inaccurate. It states “...across the dorsal bulb for MV (2nd row) are closer to BA (3rd row) than to HEX (bottom)”, while all three images are in the same row.

We are especially grateful to the reviewer for pointing out this typographical error. We have put numbers on the images and referred to these numbers in the captions instead.

9. Page 11. "...OSN inputs to the bub", where "bub" should be "bulb".

10. On page 7, "... the effect of overlapping glomeruli resembled the effect in OSNs (Fig. 2D, ...". It should be Fig. 3D instead of Fig. 2D.

11. In Fig. 4E legend, "with color scale bar a right" should be "...at right".

We have corrected these typographical errors.

Reviewer #3:

We thank the reviewer for their contributions to our manuscript and hope that the experience was a positive one.